# Measurement Time in the Evaluation of Whole-Body Vibration: The Case of Mechanized Wood Extraction with Grapple Skidder

**Roldão Carlos Andrade Lima** [1,*], **Luciano José Minette** [2], **Danilo Simões** [1], **Qüinny Soares Rocha** [1], **Ricardo Hideaki Miyajima** [1], **Gabriel Fratta Fritz** [1], **Stanley Schettino** [3], **Denise Ransolin Soranso** [4], **Glícia Silvania Pedroso Nascimento** [2], **Marlice Paes Leme Vieira** [2], **Bruno Leão Said Schettini** [2] and **Arthur Araújo Silva** [2]

[1]  School of Agriculture, São Paulo State University (UNESP), Botucatu 18610-034, Brazil
[2]  Department of Production and Mechanical Engineering/Department of Forest Engineering, Federal University of Viçosa (UFV), Viçosa 36570-900, Brazil; minette@ufv.br (L.J.M.); bruno.schettini@ufv.br (B.L.S.S.); arthur.araujo@ufv.br (A.A.S.)
[3]  Institute of Agricultural Sciences, Federal University of Minas Gerais (UFMG), Montes Claros 39404-547, Brazil; schettino@ufmg.br
[4]  Institute of Production Engineering and Management, Federal University of Itajubá (UNIFEI), Itajubá 37500-903, Brazil; denise_soranso@unifei.edu.br
*  Correspondence: roldao.carlos@outlook.com

**Abstract:** The grapple skidder is a self-propelled forestry machine that is used for the extraction of trees in wood harvesting—commonly used in full tree systems. Moving this machine can expose operators to occupational hazards of physical origin, among which whole body vibration stands out. However, the measurement of this risk agent does not have a standard measurement time, being performed for periods of approximately 30 min—disregarding the time of the daily workday. In view of this, it was analyzed whether occupational exposure to whole body vibration transmitted to grapple skidder operators using different measurement times complies with the guidelines for preventative purposes. Thus, measurements of whole-body vibration were carried out along three orthogonal axes over a period of one hour and over a daily workday of eight hours—by which were measured the daily (8 h) vibration exposure for the *l*-axis and the vibration value. The acceleration values in the three evaluated axes were higher for the daily working day, denoting the influence of the measurement time. In addition, the vibration dose value resulted in values above the action limit for both evaluations; however, the daily workload was highlighted—indicating the presence of higher vibration peaks over a longer measurement time. Thus, the assertiveness and influence of measurement times over the daily working day for whole-body vibration transmitted to grapple skidder operators is evidenced.

**Keywords:** forestry operations; wood harvesting; occupational vibration; occupational health

## 1. Introduction

The forest sector is considered of paramount importance for economic activities in Brazil. The production of cellulose and fiber boards are two of the main items derived from Eucalyptus planted forests, which are used as raw materials for the manufacture of a wide variety of products. Consequently, there has been an increase in the level of the mechanization of forestry operations, allowing for the maximization of activities.

The mechanization of these activities from a technical, economic, environmental, and ergonomic point of view is essential for providing safety information for planning and decision making in planted forests [1]. Wood harvesting is the stage of the forestry process that most benefits from this advance, using self-propelled forestry machines for cutting, extracting, and processing wood, ensuring increased productivity and optimizing processes [2–4].

On the other hand, the frequent use of these machines in wood harvesting has caused the emergence of ergonomic adversities—whether biomechanical problems, repetitive strain injuries, work-related musculoskeletal disorders, noise, or vibrations, among others [5–7]. The physical aspect derived from mechanical vibration stands out among the adversities to which self-propelled forest machine operators are exposed.

In this perspective, such a vibration can be defined as an oscillatory movement that repeats itself after a time interval, which occurs due to unbalanced forces from rotating components and the alternating movements of machines [8,9].

Whole-body vibration stands out among the main types of vibration, characterized by the incidence of this agent on the bodies of operators along three orthogonal axes that occur in work activities that demand the execution of tasks—for example, in the sitting position [10,11].

Upadhyay et al. [12] points out that the risk factors for exposure to whole-body vibration can be classified into four categories: personal factors (age and weight of the operator); factors related to ergonomics (awkward posture and seat design); machine-related factors (vehicle age and load characteristics); and operation-related factors (machine speed, workforce, and haul road).

The effect of whole-body vibration can be aggravated in systems that use the grapple skidder as the self-propelled forest machine responsible for the extraction of bundles of trees. These machines carry out the extraction in the form of a drag—thus requiring a greater driving force to carry out the activity. In addition, the operation takes place inside the forest stand, so the grapple skidder travels over forest residues, stumps, and steep and stony areas—potentially exposing operators to occupational hazards [13–15].

Research addressing whole-body vibration in self-propelled forestry machines in Brazil is limited, being limited to studies carried out by Almeida, Abrahão, and Tereso [16], Lima et al. [17], Martins et al. [18], Cazani et al. [15], and Santos et al. [19]. Furthermore, these surveys were not carried out based on the daily workday—that is, they considered the evaluations for periods of approximately 30 min, based on the International Organization for Standardization ISO 2631-5 [20] and the Ergonomic Guidelines for Forest Machines [21].

Thus, it is noteworthy that these guidelines do not prescribe a standard time for the collection of data on whole-body vibrations for self-propelled forest machines; therefore, these periods may be considered short, resulting in the underestimation of the resulting vibration dose values—or rather, operational elements that could exert important oscillatory movements may not be contemplated during data analysis.

Zhao and Schindler [22], Huang and Zhang [23], and Delcor et al. [24] also indicated that standards do not always address the ergonomic safety issues of various activities that operators may perform in different work sectors; thus, a standard's weighting coefficients should be adapted to each environment or to each means of transport to which the standard applies.

Other limitations of studies on whole-body vibration have also been highlighted by Kowalski and Zając [25]; the number of evaluated operators, the evaluation time, and the scope of all movements performed—in addition to the use of unsuitable equipment such as industrial accelerometers in the evaluation of human body vibration—can generate results that do not represent the reality of the operation.

Although research on whole-body vibration in wood harvesting machines is incipient, the topic has already been widely addressed in machines used in silviculture and agriculture. Santos et al. [26], Singh et al. [27], and Singh et al. [28] have reported in their research that soil preparation activities, which are the initial stage of the production process, are also one of the stages that most expose operators to whole body vibration due to the rupture of compacted soil layers. With regards to this, Singh et al. [29] reported that prolonged exposure to this risk agent during this operation can result in compressive stress on the lumbar spine of workers.

In addition to this, other silvicultural activities—or even agricultural activities—that require the movement of agricultural tractors through the production area can expose

workers to the same risks. These activities include—but are not limited to—crop planting, fertilization, liming, irrigation, and pesticide application [30–32]. In this sense, the preservation of workers' occupational health becomes a relative aspect that should take into account the continuity of production in operations that use machines in agricultural and forestry areas.

Due to the importance that occupational exposure to whole-body vibration can have on operators of self-propelled forest machines used in wood harvesting, especially grapple skidders, frequent evaluations are justified—especially those that contemplate the daily work of operators—allowing the observation of all operational elements, whether cyclical or not, to support possible ergonomic improvements. Thus, here it was analyzed whether occupational exposure to whole body vibration transmitted to grapple skidder operators, with different measurement times, is in accordance with the guidelines for preventative purposes.

## 2. Materials and Methods

Initially, the research was submitted and approved by the Research Ethics Committee (CEP) of the São Paulo State University, School of Medicine, Botucatu, Brazil, according to the Certificate of Presentation of Ethical Appreciation number 25739519.1.0000.5411.

Thus, the evaluated operators participated voluntarily, receiving written clarifications about the methodology and research objectives by reading and signing the Free and Informed Consent Form, in compliance with Resolution No. 466/2012 CNS/MS from the National Research Ethics Committee of the Brazilian Ministry of Health [33].

The research was carried out in areas of mechanized wood harvesting in Eucalyptus planted forests. Thus, the planted forest was spaced 3 m × 2 m, aged 19.87 years, in the second rotation, with an average diameter at breast height (DBH) of 16.90 cm, average height of 26.2 m, and average individual volume (AIV) of 0.38 m$^3$, conducted under high-forest conditions.

The study was carried out in a single forest block with a flat relief class and a slope of 0.0% to 3.0% [34]. This block had forest residues disposed on the ground, which are a result of cutting the wood. These residues are composed of leaves, branches, bark, and tree stumps. Block stumps have a standard height of ≤4 cm in order to minimize damage to the self-propelled forest machine wheelsets.

The climate of the study region was classified by the Köppen–Geiger methodology as a Cfa humid subtropical climate [35]. The soil was classified as moderate or weak and as dystrophic Quatzarene Orthic Argisol or Neosol [36]. The harvesting system employed in the study area is the full tree; therefore, the dragging of the tree bundles to the edges of the forest roads was carried out by means of grapple skidders.

The evaluated grapple skidder was made by John Deere, model 948 L, with an engine power of 210 kW, grapple area of 2.07 m$^2$, and with 370 h of accumulated use (Figure 1), which represents the useful life of the machine—that is, it is a practically new machine. In this sense, the grapple skidder was already in continuous use at the company, with the original factory cushioning system and machine seat.

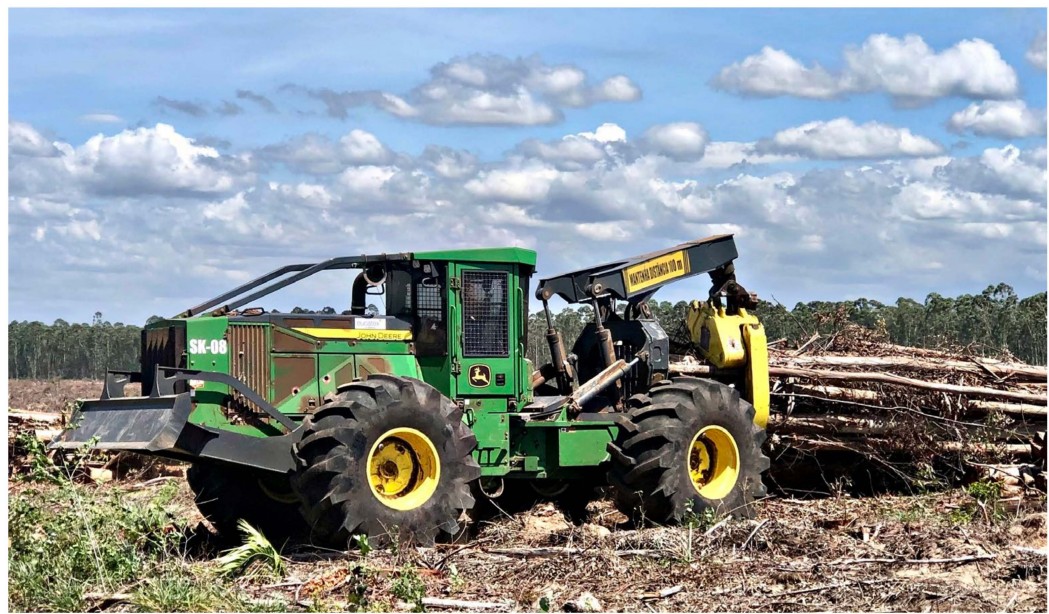

**Figure 1.** Grapple skidder evaluated, made by John Deere (Moline, IL, USA), model 948 L, with an engine power of 210 kW, grapple area of 2.07 m², and 370 h of accumulated use.

Two grapple skidder operators were considered—both male, aged 30 and 31, with a body mass of 84 and 87 kg, respectively. Both workers have five years of experience as grapple skidder operators, working eight hours a day.

The measurement of occupational exposure to whole-body vibration was based on the International Organization for Standardization ISO 2631-5 [20]; this follows three directions of a system of orthogonal coordinate axes, according to the ISO 8041 standard [37], with measurements on the $x$, $y$, and $z$ axes—where $x$ corresponds to the anteroposterior vibration transmitted to the operator's body, $y$ corresponds to the side-to-side vibration, and $z$ corresponds to the longitudinal vibration.

The collection of whole-body vibration data for the mechanized extraction activity of tree bundles was measured during a period of one hour (Test A) and during the daily journey of eight hours (Test B), using a vibration meter seat pad type, model VIB008, 01dB mark (São Paulo, SP, Brazil) (Figure 2), with frequency weighting of type Wd for the $x$ and $y$-axes and Wk for the $z$-axis. The seat pad was attached to the seat of the grapple skidder, respecting the location of the axes, which captured the readings every second throughout the journey.

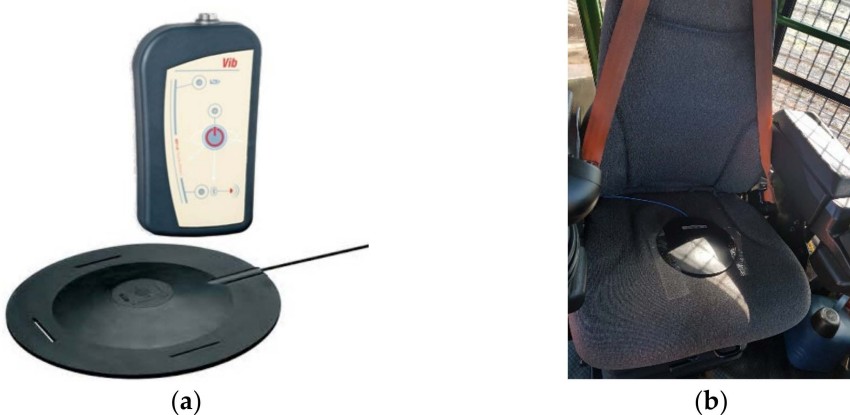

(**a**)　　　　　　　　　　　　　　　　　　　　　　　(**b**)

**Figure 2.** (**a**) Seat pad type vibration meter; (**b**) seat pad fixed to the machine seat.

According to the International Organization for Standardization ISO 2631-5 [20], the basic parameters for evaluating exposure to whole-body vibration correspond to daily (8 h) vibration exposure for the *l*-axis ($a_{wl}(8)$) and the vibration dose value (*VDV*). Thus, the total acceleration value of Test A was standardized for a period of 8 h (Equation (1)), while Test B used the actual values obtained.

$$a_{wl}(8) = \left( \frac{1}{T_0} \sum a_{wlj}^2 \times T_j \right)^{\frac{1}{2}} \tag{1}$$

where $a_{wl}(8)$ is the daily (8 h) vibration exposure for the *l*-axis, where *l* = *x*, *y* or *z* (m s$^{-2}$ rms); $T_0$ is the reference duration time in 8 h or 28,800 s; $a_{wl}$ is the global weighted acceleration on the *l*-axis, where *l* = *x*, *y* or *z* (m s$^{-2}$ rms); and *T* is the measurement duration in seconds.

The vibration dose value (*VDV*) takes into account the vibration peaks that occur during the measurement time in the three orthogonal axes (Equation (2)); however, it disregards the time of exposure to vibration.

$$VDV = k_l \left( \int_0^T \left[ a_{wl}(t)^4 \right] dt \right)^{\frac{1}{4}} \tag{2}$$

where *VDV* is the vibration dose value (m s$^{-1.75}$); $k_l$ is the multiplication factor for the *l* direction (*k* = 1.4 for *l* = *x*, *y*; *k* = 1.0 for *l* = *z*); $a_{wl}(t)$ is the weighted acceleration as a function of time between 0.5 and 60 Hz (m s$^{-2}$); and *T* is the measurement duration time in seconds.

The results of $a_{wl}(8)$ and *VDV* were analyzed for preventative purposes, using as a basis the international guidelines of the American Conference of Government Industrial Hygienists [38], which establishes the criteria for characterizing unhealthy working conditions resulting from exposure to physical and biological agents. In addition, the statistical analysis was conducted following the assumptions of Griffin [39] in a Completely Randomized Design (CRD), with two tests (A and B) and variables composed of 120 samples of accelerations in the *x*, *y*, and *z* axes.

The data were submitted to the Shapiro–Wilk test [40] and the Bartlett test [41] to verify the assumptions of normality and homoscedasticity. The Kruskal–Wallis test [42] was applied, analogous to the analysis of variance for non-parametric data, in order to verify the existence of a statistically significant difference in whole body vibration in the grapple skidder between the tests at 5.0% significance.

## 3. Results

The mean acceleration values (Table 1) indicated higher values for Test B in the three orthogonal axes, which may denote the influence of the measurement time on the vibration transmitted to the grapple skidder operators for an 8 h daily work shift. In addition, the treatments differed statistically among themselves in the three orthogonal axes, at 5.0% significance according to the Kruskal–Wallis test, which is highlighted by the *p*-value obtained being less than 0.05.

**Table 1.** Kruskal–Wallis test for the acceleration (m s$^{-2}$) transmitted to the grapple skidder operators in the three orthogonal axes over two measurement times.

| Orthogonal Axis | Mean Acceleration (Test A) | Mean Acceleration (Test B) | Coefficient of Variation | Standard Deviation | *p*-Value |
|---|---|---|---|---|---|
| *x* | 27,104.21 b | 34,975.02 a | 54.07% | 17,860.12 | $2.169 \times 10^{-5}$ |
| *y* | 10,865.29 b | 16,373.49 a | 77.94% | 11,610.11 | 0.0001253 |
| *z* | 13,448.51 b | 19,239.26 a | 64.62% | 11,498.81 | $1.466 \times 10^{-5}$ |

Means followed horizontally by different lowercase letters differ statistically at 5.0% significance, using the Kruskal–Wallis test.

The standard deviation is a measure that expresses the degree of dispersion of a data set—that is, the farther the values are from zero, the more uneven the data. Thus, the *x*-axis presents greater heterogeneity of acceleration data compared to the other evaluated orthogonal axes. In addition, the coefficient of variation—which identifies data dispersion by the relationship between the standard deviation and the arithmetic mean—identified higher values for the *y*-axis, indicating a more heterogeneous group of data in relation to the mean.

The analysis of data dispersion allowed the verification of higher values of acceleration in the *y* and *z*-axes for Test B, which can be attributed to the presence of 75% of the data in the third quartile (Figure 3). Although Test A has a greater interquartile range, 25% of the values are in the first quartile—indicating the greater proximity of the data to the zero value. Even on the *y* and *z*-axes for Test A, values equal to zero were observed, which may be due to the shorter measurement time of this analysis.

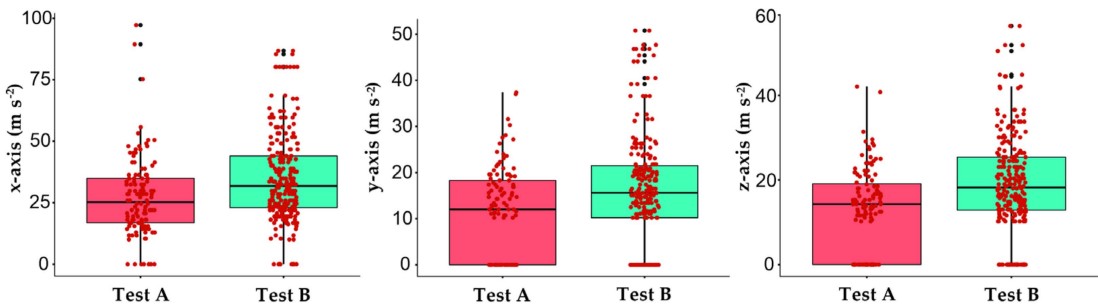

**Figure 3.** Boxplot of whole-body vibration data transmitted to grapple skidder operators in three orthogonal axes over two measurement times. Note: The red dots are the scattering of the whole-body vibration data. The black dots are the outliers.

When analyzing the values of $a_{wl}(8)$ and $VDV$ (Table 2), the highest values were found for the measurement made over the entire daily working day—above the limits recommended by the guidelines for preventative purposes. In this sense, this demonstrates the assertiveness and influence of the measurement time in the obtaining of reliable whole-body vibration values during a full shift in grapple skidder operators.

**Table 2.** Daily (8 h) vibration exposure for the $l$-axis and vibration dose value for the grapple skidder.

| Test | $a_{wl}(8)$ (m s$^{-2}$ rms) | $VDV$ (m s$^{-1.75}$) |
|---|---|---|
| A | 0.417 | 21.900 |
| B | 0.696 | 30.400 |

## 4. Discussion

Among the orthogonal axes analyzed, the *x*-axis showed the greatest discrepancy in mean acceleration between the tests—in which, Test B presented a value 23% higher than Test A, followed by the *z* (19%) and *y* (16%) axes, respectively.

Empirically, the element of the grapple skidder's operational cycle referred to as the displacement with load, caused anterior–posterior oscillations in the operators' body—as corroborated by Jack et al. [43] and Barros et al. [44], who claim that the main movements that echo inside the cabin concern the *x*-axis, which is reflected by posterior and frontal oscillations in the body of operators.

The operational cycle of the grapple skidder is divided into machine elements, which are composed of displacement without load, displacement with load, loading of wood, and unloading of wood. For displacements, the driving speed and engine power are higher—whereas for loading and unloading wood, they correspond to slow movements or movements only made by the machine's grapple [45].

Thus, it can be attributed that the grapple skidder spends most of its working day being active, that is, performing activities that demand the displacement of the machine—mainly displacement with a load of wood. This explains the steep mean acceleration values for the *x*-axis that are echoed in the machine's cab [46].

Test A presented the lowest mean acceleration on the *y*-axis—that is, the grapple skidder had smaller lateral oscillatory movements during the one-hour measurement. Sessions and Wimer [47] point out that the mobility range of this self-propelled forest machine occurs mostly in the frontal direction, which can minimize the lateral movements transmitted to operators.

Occupational exposure to too many acceleration values can damage the health of operators and, consequently, they may leave work due to occupational diseases. In this sense, whole-body vibrations can be associated with the most diverse harmful effects that occur on the health of the operator, such as fatigue, insomnia, headaches, tremors, problems in the lumbar region and neck, and reduction in the attention of operators, in addition to generating discomfort and loss of efficiency while working [48–51].

In addition to these, other regions of the body may be vulnerable to the actions of whole-body vibrations, such as bone structures, knees, muscles, and tendons [52–61]. Thus, the quantification of this exposure is a topic that stands out due to the possibility of safeguarding the health of self-propelled forest machine operators [62,63].

The mean acceleration was statistically significant ($p < 0.05$)—that is, the mean values for the daily workday in the three orthogonal axes were greater than the measurement time of one hour. This fact may indicate that the time spent performing tree extraction activities is directly related to the average values of acceleration.

Notably, the analysis of acceleration according to the orthogonal axes allows for a more careful evaluation, allowing the identification of specific points for which improvements can be made [64–66].

The coefficient of variation values indicated a greater dispersion of the data analyzed in the three evaluated axes—that is, the acceleration data were heterogeneous. In addition, the standard deviation reaffirmed the heterogeneity of the data—demonstrating values far from zero, with greater dispersion for the *x*-axis.

The analysis of the $a_{wl}(8)$ and $VDV$ values allowed for the identification of more reliable values during the 8 h measurement period. In this way, Johanning [67] and Krajnak [68] reinforced that healthcare for operators and consideration of exposure time during the working day are essential for the adoption of precautionary and preventive measures.

When comparing the values obtained to the recommended limits, we verified that the results of $a_{wl}(8)$ for Test A were below the threshold limit value and the action limit of 0.430 m s$^{-2}$ rms, following the recommendations of the American Conference of Governmental Industrial Hygienists. On the other hand, Test B presented a value above the action limit, for which Lima et al. [17] and Cazani et al. [15] indicate the need to adopt measures for the immediate control of vibration levels.

The action limit anticipates possible problems that may be caused by exposing operators close to the threshold limit value. Thus, performing activities above these limits can result in occupational diseases, such as a loss of balance, slow reflexes, increased heart rate, lack of concentration during work, blurred vision, nausea, gastritis, ulcerations, and Raynaud's syndrome, among others [69–74].

Regarding the $VDV$ values, both evaluated tests were above the threshold limit value of 17.000 m s$^{-1.75}$. However, test B stood out for its high exposure to peak vibration levels, which reached values up to 36.34% higher than test A. Thus, the evaluation during the longest measurement time identified the presence of higher vibration peaks throughout the day, emphasizing the greater reliability of measurements made over the daily working day.

Peak vibration is caused by large shocks and jolts and is identified by ISO 2631-5 as a strong risk factor for the occupational health of workers. This can be explained by the longer travel time of the grapple skidder over stumps and forest residues, since the presence of these stumps can influence the occurrence of whole-body vibration peaks transmitted to

grapple skidder operators as the machine travels throughout the field forestry [75]. Thus, the more evident the stumps, the greater their impact on the machine's tires, and—as a consequence—the vibration effect will be transmitted to the operator's body [18,76].

If we consider the three factors that quantify a vibration as the intensity and frequency of the vibration source, the isolation performance of the vibration transmission path, and the response of the structural parts [77,78], we can highlight some options for the attenuation of whole-body vibration in grapple skidder operators. Concerning the lowering of the intensity and frequency of the vibration source, we can indicate a lower height of remaining stumps and a preference for flatter terrain. Regarding the isolation performance of the vibration transmission path and the response of the structural parts as an alternative for minimizing whole-body vibration, the grapple skidder suspension systems can be improved—in addition to the seat cushions—to attenuate the emission of whole-body vibrations transmitted to workers [79–81].

In a similar study by Staněk and Mergl [82], who evaluated whole-body vibration in self-propelled forest machines of the harvester type—used in the cutting and processing of wood in cut-to-length harvesting systems—the measurement time was not mentioned, but all the operating elements of the machine were covered. Thus, in this research, the highest results for whole-body vibration were also obtained during the displacement of the machine, with an average acceleration value of 0.790 m s$^{-2}$ rms—above the limit recommended by the guidelines for preventative purposes.

In the evaluation of other machines used in wood extraction, such as the forwarder, Poje et al. [83] evaluated whole-body vibration without using a standard measurement time. In this case, they obtained maximum values $a_{wl}(8)$ of 4.75 m s$^{-2}$ rms and $VDV$ of 80.51 m s$^{-1.75}$—both above those recommended by the guidelines for preventative purposes. This result justifies the high values of whole-body vibration in wood extraction machines, regardless of the harvesting system used—whether full tree or cut-to-length, due to the displacements carried out inside the forest stand.

In operations that use the grapple skidder, Šušnjar et al. [84] evaluated whole-body vibration in two grapple skidders, with different accumulated hours of use, that worked in logging in Croatia. For this evaluation, there was no standard measurement time, covering only the machine elements of the operating cycle. Thus, an acceleration value of 1.12 ms$^2$ was obtained for the grapple skidder with the longest usage time, and 0.65 ms$^2$ was obtained for the grapple skidder with the lowest usage time. This research reaffirms the results obtained in this study, in which the whole-body vibration was greater in machines with a longer service life.

It is important to point out that surveys with whole-body vibration in self-propelled forest machines do not follow a standard measurement time, which may overestimate or underestimate the weighted values. Thus, our results indicate that measurements on these machines should be carried out during the daily workday, in order to contemplate all the important elements of the total operating cycle of the machine and to enable decision making by forest managers.

## 5. Conclusions

The acceleration over the three orthogonal axes denoted higher values for the measurement made over the eight-hour day, showing the influence of the measurement time on the accuracy of the estimation of whole-body vibrations transmitted to the grapple skidder operators.

Both evaluated measurement times showed vibration dose values higher than recommended for preventive purposes, inferring the need for the immediate adoption of measures to control this physical agent.

The measurement of whole-body vibration during the daily working day is a viable alternative for forest managers in order to guarantee the preservation of the health of self-propelled forest machine operators. This measurement guarantees more reliable and real values that may not be possible to acquire using only partial-time measurements.

**Author Contributions:** Conceptualization, R.C.A.L., L.J.M. and D.S.; methodology, Q.S.R., R.H.M. and G.F.F.; formal analysis, R.C.A.L. and Q.S.R.; investigation, R.C.A.L., Q.S.R., R.H.M. and G.F.F.; resources, L.J.M., B.L.S.S. and A.A.S.; data curation, R.C.A.L. and Q.S.R.; writing—original draft preparation, R.C.A.L., G.S.P.N. and M.P.L.V.; writing—review and editing, R.C.A.L., S.S. and D.R.S.; supervision, D.S. and L.J.M.; project administration, D.S.; funding acquisition, R.C.A.L. and L.J.M. All authors have read and agreed to the published version of the manuscript.

**Funding:** This study was financed in part by the Coordenação de Aperfeiçoamento de Pessoal de Nível Superior—Brasil (CAPES)—Finance Code 001.

**Data Availability Statement:** The data that make up the research can be obtained upon request by e-mail to the corresponding author.

**Conflicts of Interest:** The authors declare no conflict of interest.

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
