# Peer review of "Measurement Time in the Evaluation of Whole-Body Vibration: The Case of Mechanized Wood Extraction with Grapple Skidder"

_forests, doi:10.3390/f14081551_

Round 1
Reviewer 1 Report
As the authors themselves mentioned in the literature review, there are numerous factors that can influence occupational exposure to whole body vibration. The work is very well explained with regard to the comparison between a short-term sampling (generally used in scientific work, mainly given the limitations of the batteries of vibration measuring equipment) and a full-day sampling. Even so, it is necessary for the authors to BETTER CHARACTERIZE the situation to which the worker was exposed, providing more information such as, for example: skidder total hours of use (370 hours of accumulated use were reported, this is the useful life, or i.e., was it a practically new machine? Was this machine already in continuous use at the company or was it still being adapted by the workers?), height of the stumps, presence of antlers on the ground, machine cushioning system (seat, etc.) , among others. Thus, the analysis of the influence of vibration in the studied condition is fairer.
Reviewer 2 Report
Dear Authors, I had the pleasure to review your manuscript titled "Measurement time in the evaluation of whole-body vibration: the case of mechanized wood extraction with grapple skidder". This study covers the important topic.
However, there are some shortfalls in the manuscript preparation, which should be attended to, in order to improve the quality.
Please add a short overview on “state of the art in research on the area of WBV on similar machines in forestry. The best place it would be the introduction section before the aim of the paper.
line: 119: You mention that two operators took part in the research. Did both operators have the same amount of experience? Did they both weigh the same? These factors could affect the magnitude of the vibrations measured.
Table 1. It would be good to explain what mean the letters "a" and "b" after the numerical value for "Mean acceleration".
It is necessary to comment on the results in more detail in writing.
It would be good to include in the discussion the vibration values given by the manufacturer of this machine.
In the discussion it would be good to mention and compare other machines in forestry (harvesters, forwarders) and point out the magnitude of whole body vibrations. I recommend using articles that deal with whole body vibrations (For example: Staněk, L.; Mergl, V. Whole Body Vibrations during Fully Mechanised Logging. Forests 2022, 13, 630; Poje, A.; Grigolato, S.; Potočnik, I. Operator exposure to noise and whole-body vibration in a fully mechanized CTL forest harvesting system in Karst Terrain. Croat. J. For. Eng. 2019, 40, 139–150.)
I would recommend to expand the "Conclusions" with some information. In my opinion it is too short.
